# The COVID-19 Pandemic as a Global Crisis: Aspects of University Students’ Well-Being During the Quarantine in Greece

**DOI:** 10.3390/healthcare13192472

**Published:** 2025-09-29

**Authors:** Elisabeth K. Andrie, Evangelia Georgia Kostaki, Evanthia Sakellari, Sophie Leontopoulou, Areti Lagiou

**Affiliations:** 1Department of Public and Community Health, University of West Attica (UNIWA), 12243 Athens, Greece; sakellari@uniwa.gr (E.S.); alagiou@uniwa.gr (A.L.); 2Department of Hygiene, Epidemiology and Medical Statistics, Medical School, National and Kapodistrian University of Athens, 11527 Athens, Greece; ekostakh@med.uoa.gr; 3Department of Primary Education, University of Ioannina, 45500 Ioannina, Greece; sleon@uoi.gr

**Keywords:** university students, well-being, COVID-19 pandemic, online education

## Abstract

**Background:** The COVID-19 pandemic and the rapid transition to emergency online learning affected the academic and social lives of young people. This study aimed (a) to examine how university students’ well-being was influenced during the second stage of the lockdown due to the COVID-19 pandemic (2020–2021) in Greece and (b) to explore the relationship between university students’ online learning competence and well-being during the pandemic in two universities in Greece. **Methods:** A cross-sectional study was conducted during 2021 among the students at the Universities of West Attica and Ioannina. Participants completed a self-administered online questionnaire including demographics, the General Health Questionnaire-12, the Flourishing Scale Questionnaire, and an Online Education Experience Questionnaire. Focusing on the Flourishing Scale Questionnaire, factor analysis, reliability, and validity of the identified underlying factor were estimated and multivariable median regression analysis was performed to investigate determinants of well-being. **Results:** 1103 students participated, 73.6% were female, and 80.4% were aged <25 years. Analysis revealed satisfactory convergent validity and high reliability of the Flourishing Scale Questionnaire (Cronbach’s alpha coefficient = 0.88). Over 50% of the participants reported high levels of well-being. Students with greater confidence in using basic online learning tools and satisfaction with living conditions during lockdown reported well-being (*p* < 0.001). In contrast, those who reported dissatisfaction with their living conditions (*p* = 0.002) or whose living arrangements were influenced by financial concerns (*p* = 0.046) reported unwellness. **Conclusions:** The results provided useful insight into the complex interrelations between the COVID-19 pandemic quarantine, online education, and young people’s well-being, especially nowadays, when education, employment, and entertainment increasingly rely on internet-delivered modes for young people.

## 1. Introduction

In March 2020, the World Health Organization (WHO) announced the spread of severe acute respiratory syndrome coronavirus 2 (SARS-CoV-2) as a global pandemic [1]. The new coronavirus disease (COVID-19) had spread rapidly and became a public health emergency worldwide. In order to reduce viral transmission, many countries implemented a wide array of anti-epidemic measures, including physical and social distancing, wearing face masks, lockdown restrictions, and widespread closure of all educational institutions [2,3]. Over 150 countries have closed all academic institutions nationwide, including schools, colleges, and universities, affecting more than 80% of the world’s student population. Universities have moved rapidly to transition their academic programs from face-to-face to online classes [4].

COVID-19-related restrictions affected the autonomy, social development, and well-being of young people. As academic institutions were closed, education shifted from face-to-face to online modes, raising issues concerning students’ access and competence to deal with digital technology demands, as well as limitations to employment prospects. Furthermore, many universities lacked the necessary hardware and software facilities or teaching resources to support the immediate online teaching process effectively [5]. During this period, young people lost their ability to meet friends as they were restricted from their social lives, events, and usual activities. In addition, many students were obliged to return to their homes to live with their families, sacrificing their autonomy and personal independence. As a result, young people faced psychological stress, feelings of social isolation, economic uncertainty, difficulties in managing distance learning issues, as well as problems related to rapid technological changes and digital transformation of the educational process [6,7]. Due to the above challenges at this stage of their life that negatively affect decision-making and self-realization of young people, psychosocial and mental health problems of university students have been reported during the quarantine by several studies [8] concerning their development and well-being [9].

Well-being is a critical factor for an individual’s mental health. It is a dynamic and multifaceted process helping individuals live fulfilling lives. It includes various dimensions, such as emotional, psychological, and social well-being [10]. Individuals who experience high levels of well-being, classified as “flourishing”, tend to report positive functioning and strong interpersonal connections, which may protect against mental health disorders. On the other hand, individuals with impairments in some dimensions of well-being are classified as “languishing” and are likely to face higher vulnerability to depression and anxiety, representing a risk factor for mental illness [11,12]. Understanding the factors that support well-being is essential, particularly during crises like the COVID-19 pandemic, when external stressors may exacerbate psychological distress.

University students, as emerging adults, face unique mental health challenges. The abrupt shift to online education during COVID-19 disrupted their academic and social lives, creating stress, isolation, and uncertainty, affecting their mental health [13,14,15,16]. Many studies have explored the effects of COVID-19 quarantine measures on students’ mental health, resilience, and well-being all over the world. Prior research indicated that the COVID-19 pandemic, expanding over two academic years, has negatively impacted on the quality of life and well-being of university students worldwide [13]. These studies revealed increased prevalence of anxiety, stress, depression, and event-specific distress among academic students during the lockdowns [14,15,16].

Online learning competence, defined as students’ ability to effectively use digital tools and strategies to engage in learning, has emerged as a key factor influencing well-being during the pandemic. Higher competence can enhance academic self-efficacy, study engagement, and adaptive coping, whereas lower competence may contribute to frustration, disengagement, and psychological distress. Investigating this relationship is particularly relevant in contexts where online education is rapidly adopted without prior preparation [17].

In Greece, the closure of educational institutions during the quarantine and the rapid shift from face-to-face to online education modes was adopted for the first time on a massive scale by Greek authorities to counter the possible adverse effects of prolonged lockdowns on students’ learning. During the first wave of the pandemic, research on the Greek student population has focused mainly on the adverse mental health outcomes, such as anxiety, stress, and depression [18,19,20,21,22]. However, during the second stage of the lockdown, research worldwide has begun to shift toward examining more on emerging adults’ well-being as a critical protective factor for individuals’ mental health [23].

As far as we know, there is still little research on the impact of the pandemic on the Greek academic students’ well-being and its associated factors during later stages of lockdown in Greece [24,25]. Therefore, the aim of the present study was (a) to investigate how university students’ well-being was influenced during the second stage of the lockdown due to the COVID-19 pandemic (2020–2021) in Greece and (b) to explore the relationship between online learning competence and students’ well-being during the pandemic in a central and a peripheral Greek university. The findings are intended to inform strategies to support students’ well-being and to guide the development of effective preventive programs in similar crisis contexts in the future.

## 2. Materials and Methods

### 2.1. Procedure and Participants

A cross-sectional study using a self-administered questionnaire was conducted between June and October 2021. This study targeted undergraduate students at the University of West Attica and the University of Ioannina. Participants were recruited using the convenience sampling method. The study was approved by the Ethics Committees of the University of West Attica (approval code: 42768/01-06-2021) and University of Ioannina (approval code: 25852/01-06-2021). Invitation emails were sent including background information on the study, the importance of student participation, the names of the researchers, and a link to complete the survey. The email also stated that by clicking the link to complete the survey, they were providing informed consent. The survey was filled out anonymously on a voluntary basis.

### 2.2. Materials

The authors have developed a self-administered questionnaire including the following sections:

(1) Consent form.

(2) Socio-demographic information: gender, age, academic field, academic year, place of residence during the quarantine, family economic status, and parental education as proxy measures of socio-economic status.

(3) Online Education Experience Questionnaire: A 15-item questionnaire developed in order to examine students’ skills (competence in using digital tools for learning) and students’ beliefs (attitudes toward online learning effectiveness) during the COVID-19 quarantine. The items included a 5-point Likert scale ranging from “absolutely disagree” to “absolutely agree.” According to a previous validation study, this scale revealed a Cronbach’s alpha value of 0.885 [26].

(4) General Health Questionnaire (GHQ-12) [27]: A 12-item questionnaire used for the assessment of participants’ psychological symptoms and social functioning. A 4-point Likert-type scale score ranges from 1 = not at all to 4 = very often was used. Higher scores indicate higher levels of mental distress. [28,29].

(5) Flourishing Scale Questionnaire (FS): This questionnaire accesses the elements of participants’ well-being [30]. It includes eight positively worded items covering positive emotions, engagement, relationships, meaning of life, and accomplishment [31]. Each item is rated on a 7-point Likert scale (1 = strongly disagree to 7 = strongly agree), resulting in a total score ranging from 8 (lowest available score—lowest well-being) to 56 (highest available score—highest well-being or flourishing). Higher scores reflect greater psychological resources or strengths and a more positive perception of important life domains [31]. The flourishing scale has been widely accepted and has been adapted in many languages [32,33,34,35], including Greek [36]. A validated Greek version was used [36]. Cronbach’s alpha for the original scale was 0.87 [30] and for the Greek sample in previous research was 0.72 [37].

In the context of the current study, we focused on the Flourishing Scale that explored students’ well-being during the COVID-19 quarantine.

## 3. Statistical Analysis

### 3.1. Descriptive Statistics

Categorical variables were described using frequencies and percentages, and continuous variables using mean and standard deviation (SD). All statistical analyses were carried out with the use of the Stata 14.2 program [38], apart from factor analysis which was performed using the SPSS 23.0 program [39]. A significance level of 0.05 was applied.

### 3.2. Factor Analysis, Reliability, and Validity

Given that the sample of the current study had different characteristics compared to the one used in the original validation, factor structure, reliability, and validity were assessed again. Factor analysis was used to group the variables reflecting the eight items of the Flourishing Scale Questionnaire into factors. The Kaiser–Meyer–Olkin (KMO) measure of sampling adequacy and Bartlett’s test of sphericity were used to check if the data were appropriate for factor analysis and if the interitem correlations were sufficient (cut-off point for the KMO = 0.50). The principal component method using the covariance matrix was applied. We used the eigenvalue (>1.0) and the scree test to define the number of factors that would be retained [39,40].

Cronbach’s alpha coefficient was used as an indicator of reliability (cut-off point = 0.60) [41]. The factor’s convergent validity was tested by obtaining the correlations between the item score and the score of the factor to which the item belonged after deleting this item (corrected item–total correlation).

### 3.3. Median Regression Analysis

A new variable was generated by adding the scores of the eight items of the Flourishing Scale Questionnaire defining the factor. Univariable and multivariable median regression analyses were performed using the factor as the outcome variable. Median regression analysis was used because key assumptions of linear regression were not met. Only variables with a *p*-value < 0.10 in the univariable analysis were included in the multivariable model. To account for potential confounding, age, gender, nationality, decision concerning the participants’ place of residence during quarantine, satisfaction of the participants’ living conditions, family’s socio-economic status, and participants’ scores on skills and beliefs on online learning were chosen as possible explanatory variables in the multivariable model used to identify factors associated with well-being. The multivariable model was run on 1103 observations. Adjusted coefficients and 95% confidence intervals (CI) were reported.

## 4. Results

In total, 1103 students were recruited in the study. Respondent characteristics are presented in Table 1. The majority of the participants (73.6%) were females and approximately 80.4% were aged 25 years old or younger. Nearly all participants (94.3%) were born in Greece and (95.9%) had Greek nationality. The sample included individuals majoring in a range of academic subjects, referred to as “general sciences” (55.2%) whereas 44.5% declared studying “health sciences”. Nearly one quarter of the participants were second-year students (26.0%), followed by over fourth-year students (23.0%). Most of the students reported their subjective family’s socio-economic status as either medium (77.6%) or low (11.8%). Around one third of the participants stated that their maternal (38.8%) or paternal (35.6%) educational level was in “higher education”.

A high percentage of students (73.2%) stated that they were living in the place of their studies during the quarantine. There was a higher proportion among them (80.5%) who reported that during the quarantine they were living with their families, while 9.4% reported living alone, 1% with roommates, and 1.7% with relatives. Nearly one quarter (27.8%) of the participants stated that they had chosen their place of residence, influenced mainly by financial reasons, 20.6% by educational/academic reasons, 22.6% by personal reasons, and 7.5% by health reasons related to the pandemic. Around half of the students (54.2%) were very and 35.8% were moderately satisfied with their living situation during quarantine, while a further 6.7% were rather satisfied, and 3.3% were dissatisfied.

Table 2 lists the participants’ response for each statement evaluating their well-being during the second stage of the lockdown (2020–2021) due to the COVID-19 pandemic in Greece. Cumulative percentages for “agree” and “strongly agree” exceeding 50% in all 8 questions of the Flourishing Scale Questionnaire revealing students’ wellness condition (Table 3). Common responses were as follows: “I lead a purposeful and meaningful life” (64.8%), “My social relationships are supportive and rewarding” (60.5%), “I am engaged and interested in my daily activities” (64.7%), “I actively contribute to the happiness and well-being of others” (63.6%), “I am competent and capable in the activities that are important to me” (64.1%), “I am a good person and live a good life” (67.0%), “I am optimistic about my future” (54.5%), and “People respect me” (68.4%).

### 4.1. Factor Analysis Results, Reliability, and Validity Estimation

There were no missing values in the study data. The KMO measure (0.911 > 0.50) and Bartlett’s test of sphericity (*p*-value < 0.001) verified the appropriateness of the data for factor analysis. Factor analysis indicated that all communalities of the items of the Flourishing Scale Questionnaire were >0.30; therefore, no item removal was required (Table 3). The examination of eigenvalues and the scree test indicated the retention of one factor. The total variance explained by this factor was 55.2% (>50%).

The coefficient of reliability (Cronbach’s alpha) for all eight items of the factor was high (0.88 > 0.60), indicating high reliability (Table 3). By removing any of the items, we would not have had a considerable improvement in the factor’s Cronbach’s alpha (Table 3). All item–total correlations and corrected item–total correlations were well above the recommended value of 0.30, indicating a satisfactory convergent validity for the factor (Table 3).

### 4.2. Socio-Demographic Characteristics Associated with Well-Being

The mean value of the identified factor (which was generated by adding the scores of the eight items of the Flourishing Scale Questionnaire) was 44 (SD = 7.94), and its range was between 14 and 56.

Multivariable median regression analysis (Table 4), using the score of the factor as the outcome variable, showed that students with a greater comfort with basic skills on online learning had also a higher score in the factor indicating well-being (*p* < 0.001). Moreover, students who were very satisfied with their living conditions during quarantine also reported well-being (*p* < 0.001). In contrast, those who reported a little satisfied (*p* = 0.070) or completely dissatisfied (*p* = 0.002) with their living conditions during quarantine reported unwellness. Finally, those whose decision concerning the place of residence was mainly influenced by financial reasons reported unwellness compared to those whose decision was influenced by educational/academic reasons (*p* = 0.046). Analysis revealed no association between well-being and age, gender, nationality, family’s socio-economic status, and students’ beliefs about online learning.

## 5. Discussion

The present study tried to investigate how students’ well-being was influenced during the second stage of the lockdown due to the COVID-19 pandemic (2020–2021) in Greece and to explore the relationship between online learning competence and students’ well-being during the pandemic in a central and a peripheral university in Greece. Overall, the results showed that the majority of students reported high levels of well-being. Students with greater confidence in the use of online learning tools and with higher satisfaction with their living conditions reported well-being. In contrast, students who were dissatisfied with their living conditions or whose housing choices were influenced by financial concerns reported unwellness.

Well-being is a central goal of the United Nations sustainable development agenda [42,43] and a key area of interest for the World Health Organization (WHO). Well-being is typically associated with physical, social, and mental health. It includes multiple dimensions, such as emotional, psychological, and social well-being, and is considered a protective factor against future mental health problems. Assessing factors associated with well-being among students can help identify vulnerable groups requiring additional support [30].

In this study, the Flourishing Scale (FS) was used to assess students’ well-being, covering meaning of life, positive relationships, engagement, social contribution, competence, self-esteem, optimism, and social connections [30,31,32]. In line with previous research [33,34,35,36,37] and the original study [31], the scale demonstrated in our sample satisfactory convergent validity and high reliability (Cronbach’s alpha coefficient = 0.88).

The well-being of the university students’ population has been a significant major concern for many years before the pandemic. During the multidimensional crisis of the COVID-19 quarantine, this concern has intensified, and research on university students’ psychological well-being has become a priority in the academic field [44,45,46]. University students belong to a sensitive developmental period of their lives, identified as “emerging adulthood”. As mental health difficulties can manifest during this period, it is crucial to pay adequate attention to this age group during times of crises like the pandemic [47].

The United Nations reported that over a billion students were not physically attending school due to the closure of educational institutions worldwide [42]. In Greece, all academic institutions’ closure was announced on 10 March of 2020 [3], and the Greek educational system was abruptly confronted with online education. Most Greek students, who were unprepared for this sudden change, faced challenges in adjusting to online learning and new remotely delivered courses. Universities had to act quickly to support students’ education remotely [48]. The sudden transition affected students not only academically, due to campus closures, changes in assessments, and limited academic support, but socially as well, due to restrictions on social interactions and physical activities, which created feelings of isolation and disconnectedness from family and peers [49,50,51,52].

Besides the general uncertainty, the psychological trauma, the bereavement, or financial instability caused by COVID-19 pandemic, the quarantine posed more significant challenges for university students, affecting their engagement for life, dreams attainment, career prospects, academic progression, and general satisfaction [4,53]. Perhaps most concerning was the potential long-term impact which might cause lasting harm to students’ health and well-being, education, and quality of life [54].

Longitudinal studies conducted on university students during the pandemic indicated a significant decrease in young people’s well-being during the early stages of quarantine, particularly when lockdown measures were very strict [55,56]. However, later research did not show consistent findings. Studies conducted later in the pandemic indicated various patterns of well-being. Some studies reported improvements [57] or no changes [58], suggesting that as the pandemic progressed, some students adapted and coped better than expected, whereas other studies indicated further deterioration [59,60]. It remains uncertain whether mental health issues observed early in the quarantine persisted, and whether lockdown measures had a direct negative effect on young people’s well-being through the pandemic [61]. Our study indicated that during the second stage of the pandemic, university students demonstrated elevated levels of flourishing, revealing students’ wellness condition during quarantine.

Our results indicated that students’ well-being during the pandemic was not influenced by demographic factors such as age, gender, or socio-economic status, but rather by environmental and experiential factors like living conditions and digital learning competence. Specifically, our findings suggested that during the second stage of the lockdown, satisfaction with students’ living conditions during quarantine was positively associated with well-being. As far as we know, this is the first study in Greece indicating that those students who were little satisfied or completely dissatisfied with their living conditions during quarantine reported unwellness. Place of residence also has played an important role in the level of academic acceptance and satisfaction before the pandemic [62]. In addition, there is evidence that a familiar and comfortable environment may promote wellness and can help overcome negative emotions during the pandemic, so students in more familiar surroundings are more likely to cope with difficulties better [8,63].

In addition, our results demonstrate that participants whose decision concerning the place of residence was mainly influenced by financial reasons and reported unwellness compared to those whose decision was influenced by educational/academic reasons. Consistent with our findings, previous research suggests that financial instability, along with food and housing insecurity due to the pandemic, exacerbated psychological distress among university students [64,65].

Interestingly, our study indicated that students’ well-being was associated with their comfort and competence in online learning activities. While online learning can offer increased accessibility, interaction, and flexibility in education, it is not a perfect substitute for face-to-face teaching. Students had to adapt quickly to new digital learning tools and platforms and accept unfamiliar teaching methods, which posed additional stress, particularly for those with insufficient basic digital skills or increased financial instability. The sudden shift to online education, combined with increased anxiety about the academic performance process, had a negative impact on students’ well-being [66]. Prior research has shown that positive emotions can reduce the impact of stress which improves wellness. University students who were more skilled with online learning activities were easily familiarized with digital learning tools and platforms, therefore are more likely to have better wellness [67,68].

Our findings highlight the importance of a stable and supportive environment during crises. Our study results may help us understand the impact on the students’ well-being during similar crisis contexts and will be critical to plan and support effective preventative programs in the future. Universities should focus on supporting students’ digital skills, ensuring access to resources and training for online learning. Improving students’ living conditions and housing stability can enhance well-being, particularly for those facing financial or social constraints. Institutions of higher education can promote mental health through advisory services, wellness programs, and preventive interventions tailored to students’ needs during crises.

### Limitations

Our study was conducted in two universities in Greece and included a sizable number of students from all academic years. In addition, it is one of the few studies that evaluated the impact of quarantine on the Greek students’ well-being and its determinants during the COVID-19 pandemic in Greece. Furthermore, due to the anonymity of self-reporting data, selection and reporting biases are substantially limited. Nevertheless, this study was limited by adopting an online survey which could have limited its participation by those with good internet access. Additionally, the study design did not enable a comparison with data prior to the pandemic. The participants were mainly students from two universities located in big urban Greek cities, which raises issues of generalization. Finally, the results are limited due to the cross-sectional design of the study which prevents causal inferences. Generalizations concerning the impact of online learning competence or other factors examined in the present study on well-being should be limited to this specific sample and the context of the study. However, this study provides important aspects about well-being in university students during quarantine, as they capture specific socio-psychological dimensions of students and highlight potential determinants of well-being. Longitudinal studies are required to assess the long-term impact of the changed study conditions on the mental well-being of Greek university students.

## 6. Conclusions

Our findings showed that academic students in two Greek universities reported elevated levels of well-being during the second stage of the lockdown due to the COVID-19 pandemic. Satisfaction with living conditions was positively associated with well-being, highlighting the importance of a stable and supportive environment during crises. Students with greater confidence in the use of online learning tools reported well-being, suggesting that familiarity and competence with digital tools may reduce stress and facilitate adaptation to online learning. Online courses are expanding beyond the traditional face-to-face educational environment and becoming increasingly incorporated into academic institutions. The present study may provide knowledge for future researchers to help them steer a rationale for scholarly attention on areas not yet investigated relating to university students’ psychological well-being during pandemics or other potential crises. Additional research is necessary to examine the mental well-being of students under pandemic conditions in the long term and to foster students’ well-being by ensuring health-promoting study conditions. Institutions of higher education have the potential to promote the well-being of their members and provide their services to the local community in the areas of advisory, guidance, and direction through the planning and implementation of health promotion programs.

## Figures and Tables

**Table 1 healthcare-13-02472-t001:** Characteristics of the study population.

Characteristic	Number (1103)	Percentage (%)
**Gender**		
Male	287	26
Female	812	73.6
Other	4	0.4
**Age (years)**		
18–21	626	56.8
22–25	282	25.6
26–29	48	4.3
>30	147	13.3
**Born in Greece**		
Yes	1040	94.3
No	63	5.7
**Nationality**		
Greek	1058	95.9
Other	45	4.1
**University of studies**		
University of West Attica	1005	91.1
University of Ioannina	98	8.9
**Department**		
Health sciences	491	44.5
Other	609	55.2
Unknown	3	0.3
**Education level**		
1st	86	7.8
2nd	287	26
3rd	237	21.5
4th	239	21.7
>4th	254	23
**In your department of study, the courses take place:**		
Online	698	63.3
Both	130	11.8
**Are you staying at your place of study during quarantine?**		
Yes	808	73.2
No	295	26.8
**If YES (you live at your place of study during quarantine), you are staying: (*n* = 808)**		
With family	650	80.5
Alone	76	9.4
With roommate(s)	8	1
With relatives	14	1.7
In a student residence	2	0.2
With your partner	50	6.2
Other	6	0.8
Unknown	2	0.2
**If NOT (you do not live at your place of study during quarantine), you are staying: (*n* = 295)**		
With family	253	85.8
Alone	13	4.4
With roommate(s)	2	0.7
With relatives	1	0.3
In a student residence	1	0.3
With your partner	16	5.4
Other	3	1
Unknown	6	2.1
**This semester, the decision concerning your place of residence is mainly influenced by:**		
Educational/academic reasons	227	20.6
Health reasons (related to the pandemic)	83	7.5
Financial reasons	307	27.8
Social reasons	46	4.2
Personal reasons	249	22.6
Other	191	17.3
**Are you satisfied with your living conditions** **?**		
Not at all	36	3.3
A little	74	6.7
Moderately	395	35.8
Very	598	54.2
**If you are a 2nd year student or above, did you live at your place of study before lockdown? (*n* = 1.017)**		
Yes	808	79.5
No	101	9.9
Unknown	108	10.6
**How would you describe your family’s socio-economic status?**		
High	43	3.9
Medium	856	77.6
Low	130	11.8
I would prefer not to answer	74	6.7
**Maternal educational level:**		
Primary education	204	18.5
Secondary education	**387**	**35.1**
Higher education	428	38.8
Master’s/Ph.D	84	7.6
**Paternal educational level:**		
Primary education	254	23.1
Secondary education	385	34.9
Higher education	393	35.6
Master’s/Ph.D.	71	6.4

**Table 2 healthcare-13-02472-t002:** Participants responses on the eight items of the Flourishing Scale Questionnaire.

Question	Number	Percentage (%)
**I lead a purposeful and meaningful life.**		
Strongly agree	280	25.4
Agree	435	39.4
Slightly agree	138	12.5
Neither agree nor disagree	122	11.1
Slightly disagree	63	5.7
Disagree	46	4.2
Strongly disagree	19	1.7
**My social relationships are supportive and rewarding.**		
Strongly agree	210	19
Agree	457	41.5
Slightly agree	201	18.2
Neither agree nor disagree	127	11.5
Slightly disagree	57	5.2
Disagree	41	3.7
Strongly disagree	10	0.9
**I am engaged and interested in my daily activities**		
Strongly agree	268	24.3
Agree	446	40.4
Slightly agree	173	15.7
Neither agree nor disagree	105	9.5
Slightly disagree	62	5.6
Disagree	38	3.5
Strongly disagree	11	1
**I actively contribute to the happiness and well-being of others.**		
Strongly agree	213	19.3
Agree	489	44.3
Slightly agree	200	18.2
Neither agree nor disagree	137	12.4
Slightly disagree	28	2.6
Disagree	29	2.6
Strongly disagree	7	0.6
**I am competent and capable in the activities that are important to me.**		
Strongly agree	243	22
Agree	464	42.1
Slightly agree	192	17.4
Neither agree nor disagree	96	8.7
Slightly disagree	61	5.5
Disagree	35	3.2
Strongly disagree	12	1.1
**I am a good person and live a good life.**		
Strongly agree	239	21.7
Agree	500	45.3
Slightly agree	162	14.7
Neither agree nor disagree	144	13.1
Slightly disagree	28	2.5
Disagree	22	2
Strongly disagree	8	0.7
**I am optimistic about my future.**		
Strongly agree	254	23
Agree	347	31.5
Slightly agree	188	17
Neither agree nor disagree	150	13.6
Slightly disagree	77	7
Disagree	53	4.8
Strongly disagree	34	3.1
**People respect me.**		
Strongly agree	185	16.8
Agree	569	51.6
Slightly agree	160	14.5
Neither agree nor disagree	136	12.3
Slightly disagree	41	3.7
Disagree	9	0.8
Strongly disagree	3	0.3

**Table 3 healthcare-13-02472-t003:** Descriptive statistics of the eight items of the Flourishing Scale Questionnaire. Factor and item analysis results. Reliability statistics.

		Descriptives			Item Analysis			Factor Analysis
Q	Item	Mean	Standard Deviation	Percentage (%) of Subjects Responding “Agree” and “Strongly Agree”	Item–Total ^1^	Corrected Item–Total ^2^	Cronbach’s Alpha If Item Deleted ^3^	Communalities
1	I lead a purposeful and meaningful life	5.48	1.5	64.8	0.783	0.693	0.865	0.643
2	My social relationships are supportive and rewarding	5.43	1.3	60.5	0.704	0.602	0.874	0.470
3	I am engaged and interested in my daily activities	5.54	1.4	64.7	0.787	0.705	0.864	0.631
4	I actively contribute to the happiness and well-being of others	5.55	1.2	63.6	0.670	0.571	0.877	0.417
5	I am competent and capable in the activities that are important to me	5.52	1.3	64.1	0.780	0.699	0.864	0.611
6	I am a good person and live a good life	5.62	1.2	67.0	0.802	0.736	0.862	0.635
7	I am optimistic about my future	5.23	1.6	54.5	0.792	0.697	0.865	0.668
8	People respect me	5.62	1.1	68.4	0.619	0.523	0.881	0.340

**^1^** Correlation of item with overall score (sum of all 8 variables used); **^2^** Correlation of item with overall score excluding the item (corrected for overlap; sum of the remaining 7 items used); **^3^** Cronbach’s alpha for all eight items = 0.88.

**Table 4 healthcare-13-02472-t004:** Multivariable median regression analysis using the sum of the scores of the eight items of the Flourishing Scale Questionnaire defining a factor as the outcome variable.

Explanatory Variable	b	t	95% CI		*p*-Value
Skills on online learning #	0.29	4.31	0.16	0.42	**<0.001**
Beliefs about online learning #	0.04	0.80	−0.05	0.12	0.425
Gender					
*** Female	-	-	-	-	-
Male	0.72	1.22	−0.43	1.86	0.222
Age (years)					
** 18–21*	-	-	-	-	-
*22–25*	0.34	0.54	−0.89	1.56	0.593
*26–29*	0.82	0.64	−1.70	3.33	0.525
*≥30*	0.11	0.13	−1.54	1.75	0.900
Nationality					
** Greek*	-	-	-	-	-
*Other*	−0.85	−0.65	−3.34	1.70	0.515
This semester, the decision concerning your place of residence is mainly influenced by:					
*** Educational/academic reasons	-	-	-	-	-
Financial reasons	−1.49	−1.99	−2.95	−0.02	**0.046**
Health reasons (related to the pandemic)	−0.54	−0.49	−2.66	1.59	0.622
Other	−0.78	−0.92	−2.44	0.88	0.355
Personal reasons	−0.71	−0.91	−2.25	0.83	0.365
Social reasons	0.33	0.24	−2.35	2.99	0.811
Are you satisfied with your living conditions?					
*** Moderately	-	-	-	-	-
A little	−1.95	−1.81	−4.06	0.16	0.070
Very	3.02	5.33	1.91	4.12	**<0.001**
Not at all	−4.70	−3.14	−7.64	−1.76	**0.002**
How would you describe your family’s socio-economic status?					
* Medium	-	-	-	-	-
High	0.24	0.18	−2.35	2.82	0.858
Low	−0.44	−0.54	−2.03	1.15	0.588
I would prefer not to answer	−0.90	−0.88	−2.91	1.11	0.381
Number of observations: 1103					

# Elements of the Online Education Experience Questionnaire; * reference category; CI = confidence interval; Bolded *p*-values represent statistically significant findings (*p* < 0.05).

## Data Availability

Anonymized data that support the findings of this study are available upon reasonable request from the corresponding author (EKA). The data are not publicly available due to their containing information that could compromise the privacy of research participants.

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
