# Peer review of "The COVID-19 Pandemic as a Global Crisis: Aspects of University Students’ Well-Being During the Quarantine in Greece"

_healthcare, 2025, doi:10.3390/healthcare13192472_

Round 1

Reviewer 1 Report

Comments and Suggestions for Authors

The study examined the wellbeing of university students in Greece during the second phase of the lockdown (2020-2021). It specifically analysed how wellbeing was influenced by the online learning experience and living conditions during the quarantine. An online questionnaire was administered to 1103 students. The results showed that students with greater confidence in the use of online tools and with a high satisfaction with their living conditions reported higher levels of wellbeing. In contrast, students who were dissatisfied with their living conditions or whose housing choices were influenced by financial concerns reported lower wellbeing.

Overall, the manuscript is well-structured, and the research addresses a relevant and current topic. The presented results are significant and provide valuable insights into understanding the pandemic’s impact on students’ wellbeing.

The following minor issues are presented for the Authors’ consideration.

  • Tables 1,2 and 4. It would be desirable to include footnotes explaining the meaning of bold numbers, as well as the acronyms used.
  • Discussion, LL 248-251. According to the STROBE Statement (https://www.strobe-statement.org/), a summary of findings should be provided at the beginning of the discussion section.
  • Discussion, LL 252-268. This section contains numerous details that appear to be more introductory than relevant to the specific findings of this study. Restructuring this text and relocating some of the information would lead to greater internal consistency in the discussion.
  • Discussion, LL 272-277. It would be beneficial to add references to these sentences. The inclusion of student perspectives during the pandemic is a valuable addition to this context (doi: 10.3390/ijerph20054071).
  • Several typos and punctuation errors were noted (e.g., LL 112, 129, 183, 187, 274 etc.), and a general review is therefore recommended.
  • Please ensure that all references in the text are correctly formatted according to the journal's guidelines.

The primary strength of this paper lies in its ability to correlate specific, and almost exclusively sample-specific, variables with mental well-being. These variables include a familiarity with online learning and a student's accommodation. This has allowed for conclusions with implications for stakeholders and policymakers, particularly in the context of public health, offering universities the opportunity to adopt policies aligned with the study's findings.

As with all scientific literature, the paper also has its weaknesses, notably its cross-sectional design, which prevents causal inferences, and the fact that the study was conducted during the pandemic, which limits the generalisability of the findings to other historical periods.

Author Response

We appreciate the reviewer's thoughtful reading of our work which undoubtedly improves our text. Detailed responses are provided below.

Following the reviewer’s suggestion, language was checked throughout the paper by a native English speaker. Many minor changes have been made.

Please note that page and line numbers refer to the new marked copy of the manuscript that is attached. Changes have been marked in red using “Track Changes”.

Reviewer 1:

Comments and Suggestions for Authors

The study examined the wellbeing of university students in Greece during the second phase of the lockdown (2020-2021). It specifically analysed how wellbeing was influenced by the online learning experience and living conditions during the quarantine. An online questionnaire was administered to 1103 students. The results showed that students with greater confidence in the use of online tools and with a high satisfaction with their living conditions reported higher levels of wellbeing. In contrast, students who were dissatisfied with their living conditions or whose housing choices were influenced by financial concerns reported lower wellbeing.

Overall, the manuscript is well-structured, and the research addresses a relevant and current topic. The presented results are significant and provide valuable insights into understanding the pandemic’s impact on students’ wellbeing.

The following minor issues are presented for the Authors’ consideration.

Tables 1,2 and 4. It would be desirable to include footnotes explaining the meaning of bold numbers, as well as the acronyms used.

Response: Thank you for this observation. The bold numbers in table 1 indicate the most frequent category for each variable. Bold numbers in table 2 indicate the highest percentage of responses per item. Bold numbers in table 4 indicate statistically significant associations (p < 0.05). In order to avoid confusion, bold lettering was removed from the tables 1 and 2. We have added explanatory footnotes at table 4, as suggested.

Discussion, LL 248-251. According to the STROBE Statement (https://www.strobe-statement.org/), a summary of findings should be provided at the beginning of the discussion section.

Response: Thank you for your comment We have now included a concise summary of the key findings at the beginning of the Discussion section. Lines 268-276

Discussion, LL 252-268. This section contains numerous details that appear to be more introductory than relevant to the specific findings of this study. Restructuring this text and relocating some of the information would lead to greater internal consistency in the discussion.

Response:, We acknowledge the Reviewer’s valuable comment and have followed this suggestion.  This section has been revised to reduce introductory elements, ensuring that the discussion is more focused on our specific findings. Certain information has been relocated to the Introduction for better consistency.  Lines 278-288

Discussion, LL 272-277. It would be beneficial to add references to these sentences. The inclusion of student perspectives during the pandemic is a valuable addition to this context (doi: 10.3390/ijerph20054071).

Response: We appreciate this suggestion and have incorporated relevant references to support these sentences. In particular, we have added the suggested citation.

Several typos and punctuation errors were noted (e.g., LL 112, 129, 183, 187, 274 etc.), and a general review is therefore recommended.

Response: We thank the Reviewer for noticing this issue. A thorough proofreading of the manuscript has been performed, and typographical as well as punctuation errors (including those in LL 112, 129, 183, 187, 274, etc.) have been corrected.

Please ensure that all references in the text are correctly formatted according to the journal's guidelines.

Response: We have carefully checked and revised the reference list to ensure compliance with the journal’s formatting requirements. All in-text citations and bibliography entries have been standardized accordingly.

Reviewer 2 Report

Comments and Suggestions for Authors

Dear authors, thank you for the opportunity to review the results of your study.
This study contributes to filling the knowledge gap on the impact of the pandemic on students' well-being, as well as its relationship with online learning skills. The study has high practical significance.
The strengths of the study are the theoretical elaboration of the issue, a large sample size (1103 students) from two Greek universities, the use of valid diagnostic tools, and multivariate statistical procedures.

At the same time, when reading the manuscript, there are a number of recommendations for the authors on how to improve it:
1. Tables 1 and 2 are very cumbersome, and therefore difficult to read. It is suggested that the authors remove the first column and place the information from it in the following columns, as separating lines for each parameter. This will reduce space and improve perception.
2. Also, in these tables, we would like to see detailed information on students' responses from the Online Education Experience Questionnaire, their comparison among students of different years, areas, as well as with different well-being. This will allow us to propose measures to improve the well-being of students by developing specific skills in the field of online learning.

3. The results should more clearly indicate the data in Table 4, clarify what the Skills and Beliefs indicators are, and what questionnaire they are based on. How were the data from the Online Education Experience Questionnaire used in the analysis?

4. In the discussion of the results, I would like to see sections for each of the two hypotheses and change the presentation accordingly.

5. Since the objective of the study states practical significance in the form of preventive measures for students, I would like to see the practical recommendations of the authors based on the results of the study.

6. In the Conclusions, I would like to see the results obtained more clearly for each of the two hypotheses.
In this regard, the manuscript requires minor revision before publication.
Best regards, reviewer

Author Response

Dear authors, thank you for the opportunity to review the results of your study.

This study contributes to filling the knowledge gap on the impact of the pandemic on students' well-being, as well as its relationship with online learning skills. The study has high practical significance.

The strengths of the study are the theoretical elaboration of the issue, a large sample size (1103 students) from two Greek universities, the use of valid diagnostic tools, and multivariate statistical procedures. At the same time, when reading the manuscript, there are a number of recommendations for the authors on how to improve it:

We would like to sincerely thank Reviewer 2 for the constructive and insightful comments on our manuscript. We greatly appreciate the recognition of the study’s practical significance, as well as the positive evaluation of its methodological strengths. Below we address each recommendation in detail:

  1. Tables 1 and 2 are very cumbersome, and therefore difficult to read. It is suggested that the authors remove the first column and place the information from it in the following columns, as separating lines for each parameter. This will reduce space and improve perception.

Response: Tables 1 and 2 have been reformatted, as suggested.

  1. Also, in these tables, we would like to see detailed information on students' responses from the Online Education Experience Questionnaire, their comparison among students of different years, areas, as well as with different well-being. This will allow us to propose measures to improve the well-being of students by developing specific skills in the field of online learning.

Response:  Students' responses from the Online Education Experience Questionnaire, their comparison among students of different years and areas have already been presented in detail in a previous publication from our team (https://www.itmedicalteam.pl/health-science.html). Regarding the existence of a potential association between the students’ responses from the Online Education Experience Questionnaire (that examined students’ skills and beliefs on online learning) and their responses from the Flourishing Scale Questionnaire (that examined students’ well-being), this has been tested with the multivariable median regression model presented in Table 4. Analysis showed that students with greater comfort with basic skills on online learning had also a higher score in well-being.

  1. The results should more clearly indicate the data in Table 4, clarify what the Skills and Beliefs indicators are, and what questionnaire they are based on. How were the data from the Online Education Experience Questionnaire used in the analysis?

Response: Online Education Experience Questionnaire is a 15-item questionnaire developed in order to examine students’ skills (competence in using digital tools for learning) and students' beliefs (attitudes toward online learning effectiveness) during the COVID-19 quarantine. We have also added this information in the methods section. Analyses based on the students’ responses from the Online Education Experience Questionnaire have been presented in detail in a previous publication from our team (https://www.itmedicalteam.pl/health-science.html). These two elements of the Online Education Experience Questionnaire (skills and beliefs on online learning) were used as independent variables in the multivariable median regression analysis presented in Table 4 to check for a potential association between them and students’ well-being. A clarification about skills and beliefs indicators was added in Table 4.

  1. In the discussion of the results, I would like to see sections for each of the two hypotheses and change the presentation accordingly.

Response: In line with the Reviewer’s recommendation, the Discussion has been reorganized addressing each of the two hypotheses. Lines 268-276

  1. Since the objective of the study states practical significance in the form of preventive measures for students, I would like to see the practical recommendations of the authors based on the results of the study.

Response:  We acknowledge the importance of translating our findings into practical implications. A new subsection has been added to the Discussion, where we propose preventive measures and specific skill development strategies to support students’ well-being in the context of online learning. Lines 355-363

  1. In the Conclusions, I would like to see the results obtained more clearly for each of the two hypotheses.

Response: We have revised the Conclusions to more explicitly summarize the findings in relation to each of the two hypotheses. Lines 387-391

Reviewer 3 Report

Comments and Suggestions for Authors

I would like to thank the editors and authors for the opportunity to review the article “The COVID-19 pandemic as a global crisis: Aspects of University students’ wellbeing during the quarantine in Greece” 

Please find attached the file with several comments and suggestions for the authors to consider, in order to improve the quality and clarity of the article.

The manuscript requires minor revisions. 
Kind regards. 

Author Response

I would like to thank the editors and authors for the opportunity to review the article “The COVID-19 pandemic as a global crisis: Aspects of University students’ wellbeing during the quarantine in Greece”

Please find attached the file with several comments and suggestions for the authors to consider, in order to improve the quality and clarity of the article.

The manuscript requires minor revisions.

We would like to sincerely thank Reviewer 3 for the constructive feedback and valuable recommendations. We greatly appreciate the recognition of the study’s contribution, and we have carefully revised the manuscript according to the suggestions.

Please note that page and line numbers refer to the new marked copy of the manuscript that is attached. Changes have been marked in red using “Track Changes”.

Our detailed responses are presented below:

Abstract

In the abstract, under the methodology section, you could mention that the research is cross-sectional.

Response:  We thank the Reviewer for this useful observation. The methodology section of the Abstract has been revised to explicitly state that the study employed a cross-sectional design.

Introduction

1.In lines 86–88, you mention that “However, during the second stage of the lockdown, research worldwide have begun to shift toward examining more on emerging adults’ wellbeing as a critical protective factor for individuals’ mental health [22]”. It would be useful to discuss further, to expand the reasons why adults’ well-being as a critical protective factor for individuals’ mental health and how this is connected to the COVID-19 pandemic. This could be better linked to the text in lines 89–90, where you mention that ““there is still little research on the impact of the pandemic on Greek students’ well-being”, and followed by the objectives and aims of your study.

Response:  We agree that further elaboration is warranted. We have expanded this section to explain why well-being functions as a critical protective factor for mental health, particularly during the COVID-19 pandemic, and linked this discussion to the gap regarding Greek students’ well-being.  Lines 86-109

  1. In lines 91, 93-94, you mention that “Therefore, the aim of the present study was…to explore the relationship between online learning competence and students’ well-being during the pandemic…” You could have elaborated more in the introduction on online learning competence and students’ well-being during the pandemic, before making it a research objective to investigate. For instance you could note, how the sudden shift to online learning affected their academic self-efficacy, study engagement, and overall mental health.

Response: In response to this comment, we have added a more detailed discussion on the relationship between online learning competence, academic self-efficacy, engagement, and mental health, before introducing our research objectives.  Lines 90-104

Materials and Methods

  1. In lines 128, regarding the Flourishing Scale (FS), you could provide more information, whether it has been translated and validated into Greek. If a validated Greek version exists consider to cite the reference in the references section, as well as to check if ref. [31, 32] pertain to a Greek sample (lines 135-136).

Response: We have clarified whether a validated Greek version of the Flourishing Scale exists and included the appropriate reference in the References section. References [36, 37] have been carefully checked to ensure that they pertain to Greek samples.

  1. Although, the aims of the study focus on the impact of the lockdown on well-being and its relationship with online learning competence, a factor analysis was performed to check and confirm the reliability and validity of the measurement tool within the sample. Could you further analyze and expand why you included this step in your methodology and how it connects to the primary research outcomes and the subsequent statistical analyses you conducted, and if this was intended from the beginning, consider stating it as a secondary aim of the study.

Response: Given that reliability (Cronbach’s alpha) can vary across samples depending on sample characteristics (age, gender, education, etc.) and that in the case of differences among the samples, the factor structure is good to be assessed again, the reliability and validity of the instrument were re-assessed and a factor analysis was performed to verify the existence of an underling factor also in this dataset. Our sample has different characteristics compared to those studied in the previous Greek studies. Specifically, the study of Stalikas, et al (original validation study) was conducted on Greek adults of the general population with a mean age of 35.5 years. Leontopoulou’s study was focused only on emerging adults with most of them aged between 18 and 21 years (85.1%) and studying in Education departments (60.3%) of different Greek higher education Institutions. Our study was focused on adult students with only half of them being in the age group of 18-21 years, almost 15% of them aged more than 30 years, half of them studying in Health sciences departments, and the vast majority of them were studying at the same University in Athens (University of West Attica, 91.1%). The following clarification was added in the methods section: “Given that the sample of the current study had different characteristics compared to the one used in the original validation, factor structure, reliability and validity were assessed again.”.

Regarding the connection with the subsequent statistical analyses conducted, factor analysis verified the existence of an underling factor and as a result a new variable was generated by adding the scores of the 8 items of the Flourishing Scale Questionnaire defining the factor. This variable was used as a dependent variable in the statistical models (representing well-being) to assess for potential associations between well-being and other demographic characteristics and participants’ scores on skills and beliefs on online learning. You may check the “median regression analysis” paragraph in the methods section for more details on theses analyses.

  1. It would be helpful if the authors clarified why they chose to perform median regression, instead of, for instance, standard linear regression in order to examine the variables related to well-being (lines 159, 166). Since this was the appropriate statistical choice to answer the primary objectives, a brief explanation behind the methodological rationale would help readers to understand and at the same time strengthens the analysis.

Response: We used median regression analysis due to the violation of a key assumption of linear regression analysis which was that residuals were not normally distributed basically because the dependent variable was bounded. The following text was added in the methods section: “Median regression analysis was used because key assumptions of linear regression were not met.”.

  1. To strengthen the methodology, particularly about multivariate model, and also due to the fact that the research is cross-sectional, it would be useful to clarify whether in your analysis you took into account the potential impact of confounding factors and you have adjusted for these variables. If such adjustment has not been clearly performed, this limitation should be acknowledged and discussed in the limitations section.

Response: Age, gender, nationality, decision concerning the participants’ place of residence during quarantine, satisfaction of the participants’ living conditions, family's socio-economic status, and participants’ scores on skills and beliefs on online learning, were chosen as possible explanatory variables in the final multivariable model used to identify factors associated with well-being. All these variables were simultaneously included in the model to account for potential confounding. Several clarifications were made in the “median regression analysis” paragraph in the methods section.

Results

The results section primarily focuses on descriptive statistics, the factor analysis of the Flourishing Scale, and the median regression analysis.

  1. Although the study used the General Health Questionnaire (GHQ-12) [26] for the assessment of participants’ psychological symptoms and social functioning, it is not clear how these results are used, especially in the factor analysis model or the regression models. You might consider reporting how psychological symptoms and social functioning were assessed and whether these were statistically related to students' well-being. Clarifying how this tool was used in detail either in the results section or in the discussion section would help the reader and enhance the completeness of the findings.

Response: Students’ responses on the General Health Questionnaire (GHQ-12) had been collected but this data was not included in the current study (neither in the factor analysis nor the regression models). Here we focused only on the Flourishing Scale, and we aimed to explore students' well-being during COVID-19 quarantine and its relationship with online learning competence. We sincerely thank the reviewer for the valuable suggestion! We plan to clean, prepare and analyze data collected using the GHQ-12 instrument to assess participants’ psychological symptoms and social functioning, and we will consider investigating if these are related to students' well-being.

Discussion

The findings in discussion are contextualized within the existing literature. However, certain points could benefit from further development.

  1. In line 249, it is mentioned "with psychological aspects of their wellbeing". You could clarify whether this refers to the flourishing scale or the GHQ-12 or the combination of both. In the case where you are referring to psychological functioning, but different parameters are being studied (i.e., positive well-being vs. psychological distress), it would be helpful to clarify which measure is being referenced in this context.

Response: Thank you for your comment. In the current study, we have focused on the Flourishing Scale that explored students' well-being during COVID-19 quarantine. Line 149-152. We have revised this section to clarify that the statement refers to the Flourishing Scale, to avoid ambiguity and ensure consistency. Line 152-153, and 284 - 288

  1. For the Flourishing Scale, it could be further clarified whether the study cited in reference [35], or those in references [31, 32], as mentioned in lines 135–136 ("and for the Greek sample was 0.72 [31, 32]") and in line 267 ("Cronbach’s α for the Greek sample in previous research was 0.72 [35]"), refer to the validation of the scale in Greek, so that the relevant citation can be properly included in the References section.

Response: We have corrected the referencing to ensure that the citations referring to the Flourishing Scale validation in Greek are properly attributed. The appropriate references have been added and cross-checked. Lines: 148-151 and 284-288

Limitations & Conclusion

  1. In line 365, the phrase “the results are limited due to the cross-sectional design of the study” may further be expanded. For example, it could also be noted that given the cross-sectional nature of the study, it is not possible to establish causality. For instance, generalizations regarding the impact of online learning competence or other factors examined in the present study on well-being should be limited to this specific sample and this context of the study. However this study provides important aspects about well-being in university students, during quarantine, as they capture specific socio-psychological dimensions of students and highlight the correlation of factors with well-being. For future research longitudinal or experimental research designs would be necessary to confirm any potential causal relationships.

Response : In line with the Reviewer’s suggestion, we have expanded Limitations & Conclusion section to highlight that the cross-sectional nature of the study limits causal inferences. We further emphasize that results should be interpreted within the specific context of this study and note that longitudinal or experimental designs would be necessary for future research. Lines: 366-371 and 374-380
